# Metabolomics Analyses of Cotyledon and Plumule Showing the Potential Domestic Selection in Lotus Breeding

**DOI:** 10.3390/molecules26040913

**Published:** 2021-02-09

**Authors:** Huanhuan Qi, Feng Yu, Rebecca Njeri Damaris, Pingfang Yang

**Affiliations:** State Key Laboratory of Biocatalysis and Enzyme Engineering, School of Life Sciences, Hubei University, Wuhan 430062, China; qihuanhuan0911@163.com (H.Q.); yufeng@hubu.edu.cn (F.Y.); rebecca@hubu.edu.cn (R.N.D.)

**Keywords:** *Nelumbo nucifera*, lotus seed, comparative metabolomics, functional component, UPLC-ESI-MS/MS

## Abstract

Lotus (*Nelumbo nucifera*) seeds are widely consumed as functional food or herbal medicine, of which cotyledon (CL) is the main edible part, and lotus plumule (LP) is commonly utilized in traditional Chinese medicine. However, few studies have been conducted to investigate the chemical components of CL and LP in dry lotus seeds, not to mention the comparison between wild and domesticated varieties. In this study, a widely targeted metabolomics approach based on Ultra Performance Liquid Chromatography-electrospray ionization-Tandem mass spectrometry (UPLC-ESI-MS/MS) was utilized to analyze the metabolites in CL and LP of China Antique (“CA”, a wild variety) and Jianxuan-17 (“JX”, a popular cultivar). A total of 402 metabolites were identified, which included flavonoids (23.08% to 27.84%), amino acids and derivatives (14.18–16.57%), phenolic acids (11.49–12.63%), and lipids (9.14–10.95%). These metabolites were classified into ten clusters based on their organ or cultivar-specific characters. Most of these metabolites were more abundant in LP than in CL for both varieties, except for metabolites belonging to organic acids and lipids. The analysis of differentially accumulated metabolites (DAMs) demonstrated that more than 25% of metabolites detected in our study were DAMs in CL and LP comparing “JX” with “CA”, most of which were less abundant in “JX”, including 35 flavonoids in LP, 23 amino acids and derivatives in CL, 7 alkaloids in CL, and 10 nucleotides and derivatives in LP, whereas all of 11 differentially accumulated lipids in LP were more abundant in “JX”. Together with the fact that the seed yield of “JX” is much higher than that of “CA”, these results indicated that abundant metabolites, especially the functional secondary metabolites (mainly flavonoids and alkaloids), were lost during the process of breeding selection.

## 1. Introduction

Sacred lotus (*Nelumbo nucifera* Gaertn., *N. nucifera*) belonging to the *Nelumbonaceae* family [1] is a perennial aquatic angiosperm. It was domesticated in Asia about 7000 years ago [2], and has been cultivated for agricultural purposes for over 2000 years in China [3]. It has multiple-usage in ornamental, agriculture, and medicine. The cultivated lotus could be classified into flower, rhizome, and seed lotus according to the agronomic traits and usage in reality. Among them, lotus seeds are popularly consumed as functional food, to which both yield and nutrition are important traits. The yield of lotus seed mainly depends on its size and the number of seeds per seedpod. With the continuous improvement of living standards, more and more attention has been paid to people’s health and healthcare, and thus, it is important to increase nutritional value, as well as yield, in lotus seed production. 

Lotus seeds are rich in nutrients and functional ingredients, such as proteins, amino acids and derivatives, organic acids, saccharides, vitamins, flavonoids, and alkaloids [4,5,6]. Lotus seed could be consumed either when fresh or when mature. The dry matured lotus seed can be processed into a variety of foods, such as snacks, mooncake, noodles, beverages, or tea [6]. Besides the basic nutrients, the lotus seeds are also rich in bioactive compounds, including flavonoids, phenolic acids, and alkaloids. These bioactive constituents have antioxidant and other healthcare value and, hence, could be used as medicine in all sorts of diseases, such as cancer [7], obesity [8,9], inflammation [10], and infection [11]. 

Many studies have been conducted focusing on flavonoids and alkaloids [12,13,14,15,16], which are important components influencing food quality. The metabolomics approach will facilitate modern breeding and address problems associated with productivity and sustainable agriculture [17]. The metabolism-based approach has also been used for analyzing the metabolites composition of different lotus organs. Dynamic changes of metabolites during seed development based on gas chromatography-mass spectrometry (GC-MS) were analyzed, and the results revealed that 15 DAP (Days After Pollination) is the key switch point from highly active tissue to the preparation of storage tissue [18]. The metabolism-based methods were also applied to analyze the content of functional components and antioxidant activity of lotus seed and rhizome from the different growing region, with varying moisture availability, and pesticide residues in lotus seed [19,20]. However, few studies were conducted to compare the metabolites between the wild and domesticated varieties, which might be very important to show the selection trend in seed lotus breeding. Here, a widely targeted metabolome approach based on Ultra Performance Liquid Chromatography-electrospray ionization-Tandem mass spectrometry (UPLC-ESI-MS/MS) was applied to detect and compare metabolites in cotyledon and lotus plumule of wild lotus variety and popular cultivar, which identified a series of metabolites showing differential accumulation in different organs and varieties. Our results provide an unprecedented investigation in analyzing metabolites of dry lotus seeds between the wild variety and domesticated cultivar, providing the largest number of identified metabolites from lotus seeds so far, with the metabolites showing an organ-or cultivar-specific character. These provide valuable information on the metabolite composition in dry lotus seed and new sights in metabolomics-based assisting breeding.

## 2. Results

### 2.1. Metabolite Profiling of Dry Lotus Seeds

The size of lotus seeds in China Antique (“CA”) and Jianxuan-17 (“JX”) showed great variation, of which “CA” possess slender shaped seeds, while “JX” seeds are round in shape, and the overall width and length of lotus seeds in “JX” were larger than those of “CA” [21]. Here, a widely targeted metabolome approach was used to identify the metabolite profiles in cotyledon (CL) and lotus plumule (LP) in “CA” and “JX” of dry seeds. The correlations among these samples, including the quality control (QC), were detected, of which the correlations of QC samples were more than 0.99 (Appendix A). The results of PCA showed that the first principal component (PC1, 78.3%) could clearly separate LP and CL of both varieties, whereas PC2 (10.5%) could help to distinguish the different varieties of “CA” and “JX” within either LP or CL (Figure 1A). Specifically, the difference in LP was much more than that in CL between “CA” and “JX” (Figure 1A). Moreover, three biological replicates of each sample were well grouped, which was consistent with the correlation analysis (Appendix A). Together, these results demonstrated very good reproducibility and high reliability of the generated metabolomics data. 

Based on the MS analysis, a total of 402 different metabolites were identified, including 348, 338, 394, and 388 metabolites in cotyledon of “CA” (“CACL”), cotyledon of “JX” (“JXCL”), lotus plumule of “CA” (“CALP”), and lotus plumule of “JX” (“JXLP”), respectively (Figure 1B, Appendix A). Of them, 317 were identified in all the samples, 329 were commonly identified in “CACL” and “JXCL”, and 385 metabolites were commonly identified in “CALP” and “JXLP”. Five metabolites were only detected in “CA”, among which two (Isorhamnetin-3-*O*-neohesperidoside and Isorhamnetin-*O*-Hexoside-*O*-rhamnoside-*O*-rhamnoside) belonging to flavonols were only detected in CALP (Figure 1B). Six metabolites included 5-Hydroxy-l-trypotophan, *N*-Acetylaspartate, Sibiricose A6, Luteolin-*O*-eudesmic acid-*O*-gallate, and phosphoenolpyruvate were only detected in CL, and 43 metabolites only detected in LP (both in “CA” and “JX”), including primary metabolites (*N*-α-Acetyl-l-arginine, thymine, and Fumaric acid) and secondary metabolites (8 phenolic acids, 25 flavonoids, 1 lignans, 4 alkaloids, 1 terpenoids).

There were 113 different flavonoids accounting for over 23% in all the four samples, which was the major category of the total detected metabolites (Figure 1C). Amino acids and derivatives were the second abundant category, which had 58 and accounted for 14.18–16.57% in the four samples. The third category was the phenolic acids, including 50 compounds and accounting for 11.49–12.63% in the four samples. The rest categories comprised of lipids (37; 9.14–10.95%), nucleotides and derivatives (31; 7.87–8.88%), alkaloids (32; 7.40–8.12%), organic acids (19; 4.82–5.33%), and others (50; 12.44–14.37%). The categories of flavonoids, alkaloids, others, and lipids were further divided into different subcategories (Figure 1D). Nine types of flavonoids, including flavonols, flavonoid, dihydroflavone, flavonoid carbonoside, flavanols, isoflavones, anthocyanins, chalcones, and sinensetin, were detected, of which flavonols and flavonoid were the major two subcategories and contained more components in LP than in CL for both varieties. There were six different alkaloids metabolites, alkaloids, plumerane, aporphine alkaloids, isoquinoline alkaloids, phenolamine, quinoline alkaloids, and five different lipids, including free fatty acids, lysophosphatidylcholine (LPC), glycerol ester, lysophosphatidyl ethanolamine (LPE), and sphingolipids. The category of others contained saccharides and alcohols, vitamins, xanthone, and glucosinolates.

### 2.2. Metabolites Clustering

All of the detected metabolites were subjected to clustering analysis in order to clarify their organ- and variety- specific accumulation, based on which ten clusters were obtained (Figure 2A, Appendix A). The metabolites in cluster I were mainly those that accumulated in LP of “CA”, including 18 flavonoids, 9 amino acid and derivatives, and 8 nucleotides and derivatives. Cluster II contained 6 metabolites, which showed the lowest accumulation in the CL of “CA”. However, two of the six metabolites (d-Alanyl-d-Alanine and *p*-coumaroylmalic acid) in this cluster were not detected in the CL of “CA”. Cluster III contained the largest number of metabolites that were mainly accumulated in LP. In this cluster, there were 84 (44.4%) different flavonoids (Figure 2B). Eleven metabolites in cluster IV were mainly accumulated in LP of “JX” (Figure 2A and Appendix A). The metabolites in cluster V were those that mainly accumulated in CL of “JX”, which included 9 lipids and 9 others. A total of 33 metabolites in cluster VI were mainly accumulated in CL of both varieties, including amino acids and derivatives, lipids, and phenolic acids. Cluster VII contained 6 metabolites, which were highest in “JXCL” and lowest in “JXLP”. There were fifteen metabolites in cluster VIII and 42 in cluster IX. The metabolites in both clusters showed a preference accumulation in CL of “CA” (Figure 2 and Appendix A). The metabolites in cluster X showed a “CA” accumulation. Among all the clusters, cluster III was the largest one, and the following top clusters included clusters I, V, VI, and IX (Figure 2A), which were further categorized based on the chemical features of the metabolites (Figure 2B). Flavonoids were the major metabolites in both cluster I and III (Figure 2B). They also contained relative high percentage of amino acids and derivatives, nucleotides and derivatives (Figure 2B). Specifically, cluster III also contained a high percentage of alkaloids and phenolic acids, which accounted for 15.9% and 10.6% of the total metabolites in this cluster, respectively (Figure 2B). Cluster V showed JXCL preference accumulation with lipids and others as the two major metabolites (Figure 2B). Cluster VI was CL-specific with lipids, amino acids and derivatives, flavonoids, and phenolic acids being the major four groups (Figure 2B). Cluster IX was CACL specific with lipids, others, and organic acids as the top three categories (Figure 2B). 

### 2.3. Differentially Accumulated Metabolites in Two Varieties

To analyze the differentially accumulated metabolites (DAMs) among different samples in this study, the value of the variable importance in the project (VIP) and foldchange for metabolites were calculated. The metabolites with fold change ≥2 or ≤0.5 and VIP > 1 for each comparison were selected as DAMs. A total of 109, 107, 253 and 266 DAMs were identified in the comparison of “CACL” vs. “JXCL”, “CALP” vs. “JXLP”, “CACL” vs. “CALP”, and “JXCL” and “JXLP”, respectively (Figure 3A). For the comparison of “CACL” vs. “JXCL” and “CALP” vs. “JXLP”, most of the DAMs were less abundant in “JX” than in “CA” (71.6% and 72.9% in CL and LP comparison, respectively). And most of DAMs were more abundant in LP than in CL within the same variety (80.6% and 82.7% in “CA” and “JX”, respectively). Thirty-four DAMs were commonly detected in the comparison of “CACL” vs. “JXCL” and “CALP” vs. “JXLP” (Figure 3B), including 9 flavonoids, 8 amino acids and derivatives, 4 nucleotides and derivatives, and 4 alkaloids, whereas most of the DAMs (221) in the comparison of “CACL” vs. “CALP” (80.6%) were also detected in the comparison between “JXCL” vs. “JXLP” (Figure 3C).

The DAMs were classified into 12 categories and the main DAMs belonged to flavonoids, amino acids and derivatives, nucleotides and derivatives, phenolic acids and alkaloids (Table 1). Eleven of the 28 DAMs, such as genistein 7-Glucoside, apigenin-6-*C*-glucoside, genistein 8-*C*-glucoside, quercetin-3-*O*-α-l-rhamnopyranoside, and kaempferol-3-*O*-galactoside, belonging to flavonoids in “CL”, were more abundant in “JX” than in “CA”, whereas only one (cyanidin-3-*O*-glucoside) of the 36 DAMs belonging to flavonoids in LP were more abundant in “JX” than in “CA”. Three (DL-Homocysteine, d-Alanyl-d-Alanine and *S*-(methyl) glutathione) of the 26 DAMs in CL and three (1,2-*N*-Methylpipecolic acid, *N*-α-Acetyl-l-arginine and *N*-Acetyl-l-tyrosine) of the 17 DAMs in LP belonging to amino acids and derivatives were also more abundant in “JX” than in “CA”. All of the 11 DAMs (such as LysoPE 14:0, LysoPC 16:0, LysoPC 16:1, LysoPC 17:0, and LysoPC 18:0) belonging to lipids were more abundant in “JX” in the LP while two (10,16-Dihydroxy-palmitic acid and LysoPC 16:2) belonging to lipids were down-accumulated in “JX” in the CL. In the comparison of CL vs. LP, most of the DAMs, except those that belonged to organic acids and lipids, were more abundant in LP than in CL in both “CA” and “JX”, indicating that flavonoids, amino acids and derivatives, nucleotides and derivatives, and phenolic acids had higher content in LP than in CL.

### 2.4. Functional Annotation and Enrichment Analysis of DAMs

To further characterize the DAMs in the two varieties of the CL and LP tissues, Kyoto Encyclopedia of Genes and Genomes (KEGG) pathway enrichment analysis was performed on the annotated DAMs. The DAMs between “CALP” and “JXLP” were mainly enriched in flavonoid biosynthesis, purine metabolism, zeatin biosynthesis, glycerophospholipid metabolism, and isoflavonoid biosynthesis (Figure 4A), whereas the DAMs between “CACL” and “JXCL” were mainly involved in the biosynthesis of amino acids, aminoacyl-tRNA biosynthesis, 2-oxocarboxylic acid metabolism, glucosinolate biosynthesis, lysine degradation, phenylalanine metabolism, and phenylalanine, tyrosine, and tryptophan biosynthesis (Figure 4B), implying that the DAMs in LP were mainly involved in secondary metabolism and DAMs in CL were mainly involved in primary metabolism. The KEGG enrichment of DAMs in the comparison of CL and LP in “CA” and “JX” also demonstrated that the DAMs mainly involved in biosynthesis of secondary metabolites including flavonoid biosynthesis and flavone and flavonol biosynthesis (Appendix A). Flavonoids are one of the most important secondary metabolites in the plant, which are naturally occurring polyphenolic compounds and have a potential role in cardiovascular health. Seven intermediates of flavonoids biosynthesis, including pinobanksin, butin, naringenin chalcone, naringenin, eriodictyol, and dihydroquercetin, were also less accumulated in “JX” compared with “CA” (Figure 5A), which was consistent with the less accumulation of flavonoids in “JX” than in “CA”. The CL is the main storage tissue of lotus seed. Many amino acids (such as valine, leucine, lysine, arginine, tyrosine, and tryptophan) and their intermediates (such as citrate, phenylethylamine, and 4-hydroxyphenylacetate) were detected in CL of both varieties. Except for trans-cinnamate and homocysteine, most of them were less accumulated in “JX” than in “CA” (Figure 5B).

## 3. Discussion

### 3.1. Abundant Metabolites Accumulated in the Dry Lotus Seed

Lotus seed is a popular functional food and is always used to treat some ailments [6,22]. It could be eaten in the fresh status with a shorter storage period, in contrast to which dry lotus seed could be stored for a long time [23]. Numerous studies have been conducted to investigate the potential health benefits of consuming lotus seeds, and several pharmacological properties have been explored in lotus seeds [24]. Previously studies have shown that both starch and proteins were largely accumulated in lotus cotyledon during seed maturation [18,21]. However, few studies were focused on global identification of the metabolites in the lotus seeds, especially the dry lotus seed [18,24]. In this study, a widely targeted metabolome approach based on UPLC-MS/MS was applied to analyze the chemical composition of the dry seed in a wild variety “CA” and a modern cultivar “JX”. The results indicated that both CL and LP of dry lotus seeds could accumulate abundant nutritional and functional metabolites with LP accumulating more secondary metabolites than CL, which is consistent with their usage in the reality. The nutritional compounds were mainly lipids, amino acids and derivatives, nucleotides and derivatives, organic acids, and others (carbohydrates and vitamins); the functional metabolites were mainly flavonoids, alkaloids, and phenolic acids. These components might contribute to the basic nutritional and pharmacological features of lotus seed. Many functional foods, such as buckwheat, oat, and date fruit, also contain abundant functional metabolites [25,26]. Compared with buckwheat [27], lotus seeds contain fewer flavonoids but more alkaloids and phenolic acids. Although phenolic acids were also abundant in date fruit, its flavonoids and alkaloids contents are much less than lotus seed [26]. Altogether, these data showed that lotus seed is a very healthy functional food.

### 3.2. The Potential Nutritional and Medicinal Value of Lotus Dry Seed

Food products derived from plants have abundant metabolites including not only the basic primary metabolites, such as lipids and organic acids, but also secondary metabolites that are beneficial in promoting health [28]. Flavonoids and alkaloids are the two major secondary metabolites enriched in lotus seeds [13,22,29]. Of the 402 metabolites identified in our present study, over 23% belonged to flavonoids, including quercetin, isorhamnetin-3-*O*-glucoside, isorhamnetin-*O*-rutinoside, isoquercitrin, astragalin, isovitexin, isoorientin, orientin, and luteolin-7-*O*-rutinoside. These metabolites were also identified in previous composition analysis in lotus seed [13,30,31]. Quercetin, can prevent high-fat diet (HFD)-induced obesity through regulating hepatic gene expression related to lipid metabolism [32], our study detected 17 quercetin and their derivatives. Flavonoids prevent many chronic diseases, such as cardiovascular diseases and various types of cancers [33,34,35], and alkaloids have certain pharmacological activity and medicinal usage, as well [36]. Lotus seed tea has been documented to prevent ultraviolet B irradiation and loss of skin moisture [37]. It was reported that water extracts from lotus seed had a 75% whitening effect and 49% anti-wrinkle effect at 200 µg/mL [38]. A total of 13 flavonoids in lotus leaves have been separated and identified [39], thirty-eight flavonoids were identified in nine tissue that included leaves, petals, stamens, pistils and tori, lotus plumules, stalks, seed coats, seed kernels [13] and anthocyanin and non-anthocyanin flavonoid in nine different tissue of twelve lotus cultivars were also analyzed [40], of which most flavonoids have been detected in dry lotus seed. Thirty-two metabolites belonging to alkaloids were identified, which included liensinine and higenamine, that have great potential in clinical trials [41]. Moreover, abundant metabolites that belonged to amino acids and derivatives (>14.18%), lipids (>9.14%), nucleotides and derivatives (>7.87%), and organic acids (>4.82%) were detected in CL and LP, indicating the nutritional value of these tissues. Notably, the secondary metabolites, such as flavonoids, mainly accumulated in LP, and the primary metabolites mainly accumulated in CL (Figure 2), implying that CL has more potential in food value, and LP has higher potential use for medical value. 

### 3.3. The Effects of Phenotypic Selection on Metabolites Changes and Metabolomics-Assisted Breeding

The selection of target phenotypes adapting to environmental conditions and human usage directly influences the genetic structure and allele frequency, which regulates the gene expression and the final chemical products. The composition of metabolites in different species reveals the differential evolutionary process, such as in maize and rice [42]. Multi-omics studies, including genomics, transcriptome, and metabolomics in tomato, have been performed to investigate the impact of human intervention on chemical composition and demonstrated that the phenotypic selection directly or indirectly affected the metabolites component [43]. Molecular phenotypic traits of metabolites or transcripts in wheat had shown a reduction in unsaturated fatty acids and amino acids contents during domestication [44], and the comparative analysis of wild and domesticated accessions of common bean (*Phaseolus vulgaris*) using RNA sequencing technology indicated that nucleotide diversity and corresponding gene expression were decreased [45]. “CA” is a wild variety of lotus and “JX” is a popular cultivar that possesses significantly larger lotus seeds [21], which are derived from the artificial targeted selection of seed shape. Our comparative metabolomics analysis of CL and LP in “CA” and “JX” demonstrated that more than 25% metabolites were DAMs between the two cultivars in CL and LP, respectively, which included multiple metabolite categories, such as flavonoids, amino acids and derivatives, and alkaloids (Figure 3, Table 1). The DAMs in LP mainly enriched secondary metabolism-related pathways, such as flavonoids and the DAMs, in “CA” enriched in primary metabolism, such as amino acids (Figure 4). Interestingly, most of DAMs were less abundant in “JX” than in “CA”, of which only one of the 36 DAMs belonging to flavonoids, three of the 17 DAMs belonging to amino acid and derivatives, and two of the 12 DAMs that belonged to nucleotides and derivatives showed an increase in accumulation in LP of “JX”. A similar phenomenon was also observed in CL, such as DAMs, belonging to flavonoids, amino acid and derivatives, and organic acids. One metabolite (D-Glucose) involved in glycolysis was detected, but it appeared to have no significant difference across the groups. Three metabolites (fumaric acid, succinic acid, citric acid) were detected involved in the tricarboxylic acid (TCA) cycle. Fumaric acid was not detected in cotyledon but in plumule, and succinic acid and citric acid had lower content in the cotyledon of “JX” compared to the cotyledon of “CA”, demonstrating that catabolism of TCA cycle was lower in “JX” than in “CA”. These results indicated that the selection for larger lotus seeds resulted in down-accumulation of large amounts of metabolites including edible and medicinal compositions and the phenotypic-based selection of lotus seed may partly reduce the content of important metabolites required for human nourishment.

Phenotypic selection is the classical approach for breeding cultivars that adapt to the agricultural system, and molecular marker-assisted strategy has become increasingly popular in modern breeding because of the development of biological techniques. Recent studies have shown that metabolite data could be used to predict the performance of agronomic traits, ultimately assisting crop breeding [46,47]. Breeding for seed lotus has focused on improving the yield, such as increasing seed size, but consideration of the functional components has been neglected. Analysis of flavonoids in 12 lotus cultivars derived from different groups of rhizome, flower, and seed lotus demonstrate wide variations in flavonoid content [40]. Large amounts of DAMs besides flavonoids were identified in CL and LP of the two cultivars and most of them were down-accumulated in “JX”, indicating that breeding larger lotus seeds decreased the component of most metabolites. The most enriched KEEG pathway in LP was flavonoids and the primary metabolism was the most enriched KEGG pathway in CL (Figure 4 and Figure 5A,B), implying that DAMs in these pathways were the potential important target for breeding functional seed. Our results presented here provide new clues for breeding high functional components based on metabolomics-assisted strategy in the selection process of seed lotus breeding.

## 4. Materials and Methods

### 4.1. Plant Samples

“China Antique” (“CA”) is a wild lotus variety, derived from long-term seed viability (1300 years) in Liaoning Province [32], and “JianXuan-17” (“JX”) is an elite seed lotus obtained from long-term selection and breeding in Jianning, Fujian Province, both of which have been maintained by ourselves for long period of time and also used in our previous studies [21]. “JX” has superior quality, strong growth, large canopy, high seed setting rate, high and stable yield, and wide adaptability. “CA” and “JX” were cultivated in the same pool separated by a net, which provided the same growth condition and prevented the mixture of the two varieties. During flowering, both of them were self-pollinated, and the mature seeds were harvested and desiccated. Lotus pericarp of dry seeds was firstly peeled, and then the cotyledon (CL) and lotus plumule (LP) were separated. Three biological repeats of CL and LP for each cultivar with ~2 g were prepared, and twelve samples were totally collected. The sampled tissues were frozen immediately in liquid nitrogen (Wuhan SaiEr Gas Limited Company, Wuhan, China) and stored at −80 °C for further use. 

### 4.2. Metabolites Extraction

The samples were freeze-dried and ground into powder using a mixer mill (MM 400, Retsch, Mammelzen, Germany) with a zirconia bead for 1.5 min at 30 Hz. Then, 0.1 g of the powder was weighed and dissolved in 1.2 mL of 70% methanol solution, followed by incubation at 4 °C overnight, during which it was vortexed 6 times (30 s every 30 min) in total. Following centrifugation at 12,000 rpm for 10 min, the extracts were filtrated (SCAA-104, 0.22 μm pore size; ANPEL, Shanghai, China, http://www.anpel.com.cn/ (accessed on 4 January 2019).) before UPLC-MS/MS analysis (Applied Biosystems, Foster, CA, USA). For preparing quality control (QC) samples, extracts of the twelve samples were combined into one sample that was further divided into three QC samples, which were injected as every fourth experimental sample and analyzed using the same methods for the experimental samples. 

### 4.3. Metabolites IdentificationD

The UPLC-ESI-MS/MS system, including UPLC system (Shim-pack UFLC SHIMADZU CBM30A, www.shimadzu.com.cn/ (accessed on 4 January 2019)) and MS system (Applied Biosystems 4500 Q TRAP, www.appliedbiosystems.com.cn/ (accessed on 4 January 2019)) was employed to analyze the extracts. The experimental procedures were set as follows: UPLC column, Agilent SB-C18 (1.8 µm, 2.1 mm × 100 mm); the mobile phase system, pure water with 0.1% formic acid (solvent A) and acetonitrile (solvent B); gradient program, 95:5 V(A)/V(B) at 0 min, 5:95 V(A)/V(B) at 9 min and kept for 1min, 95:5 V(A)/V(B) at 11.1 min and kept for 2.9 min; temperature of column oven, 40 °C; flow rate, 0.35 mL/min; and injection volume, 4 µL. The effluent was alternatively connected to an ESI-triple quadrupole-linear ion trap (Q TRAP)-MS/MS (ESI-Q TRAP-MS/MS). 

A triple Q TRAP (API 4500 Q TRAP UPLC/MS/MS System), equipped with an ESI Turbo Ion-Spray interface, was used to acquire linear ion trap (LIT) and triple quadrupole (QQQ) scans, which was operated in positive and negative ion mode and controlled by Analyst 1.6.3 software (Applied Biosystems Company, Framingham, MA, USA). The ESI ion source was turbo spray; ESI temperature was 550 °C; ion spray voltage of positive and negative ion mode was 5500 V and 4500 V, respectively; ion source gas I (GSI), gas II (GSII), curtain gas (CUR) were set at 50, 60, and 30.0 psi, respectively; the collision induced dissociation (CAD) parameter was set as high. The 10 and 100 μmol/L polypropylene glycol solutions (Sigma-Aldrich, Milwaukee, WI, USA) were applied for instrument tuning, and QQQ and LIT modes were performed for mass calibration. QQQ scans were acquired as multi-reaction monitoring (MRM) experiments with collision gas (nitrogen) set to 5 psi. The optimized decompression potential (DP) and collision energy (CE) were used for individual MRM transitions and the specific set of MRM transitions were monitored for each period according to the metabolites eluted within this period [48] (Chen et al., 2013b).

### 4.4. Qualitative and Quantitative Analysis of Metabolites

The self-built widely targeted metabolome database MWDB (Metware biotechnology Co., Ltd. Wuhan, China, http://www.metware.cn (accessed on 4 January 2019)) was used to identify metabolites in CL and LP of “CA” and “JX”, which has been described in previous studies [43,48]. Qualitative analysis of metabolites was conducted based on the secondary spectrum information, and repeat signals of K^+^, Na^+^, NH_4_^+^, and other large molecular weight substances were removed. The quantitative analysis of metabolites was conducted according to the MRM mode using triple quadruple-bar mass spectrometry. The software of MultiQuant version 3.0.2 (AB SCIEX, Concord, ON, Canada) was used to integrate and correct the chromatographic peaks, and chromatographic peak area integrals represented the corresponding relative abundance of metabolites.

### 4.5. Statistical Analysis

The data of metabolites amount were scaled by unit variance, and then the principal component analysis (PCA) was carried out based on statistic function ‘prcomp’ in R software (R Development Core Team 2013; version 3.5.0; www.r-cran.org (accessed on 4 January 2019)) and clustering analysis of metabolites was conducted using R package ‘pheatmap’ [49]. For orthogonal signal correction and partial least squares-discriminant analysis (OPLS-DA), the data were firstly transformed by log2 function, followed by mean centering, which were further subjected to analyze score plots and permutation plots using R package ‘MetaboAnalystR’ [50]. Volcano plots were drawn through R package ‘ggplot2′ [51], and Venn diagrams were shown by TBtools software [52]. The variable importance in the project (VIP) based on OPLS-DA results and log_2_ (Fold Change) were used to screen differentially accumulated metabolites. The KEGG Compound database (http://www.kegg.jp/kegg/compound/ (accessed on 4 January 2019)) was used to annotate the identified metabolites. The annotated metabolites that mapped to pathways were subjected to metabolite sets enrichment analysis, in which the significance was determined by hypergeometric test’s *p*-values.

## Figures and Tables

**Figure 1 molecules-26-00913-f001:**
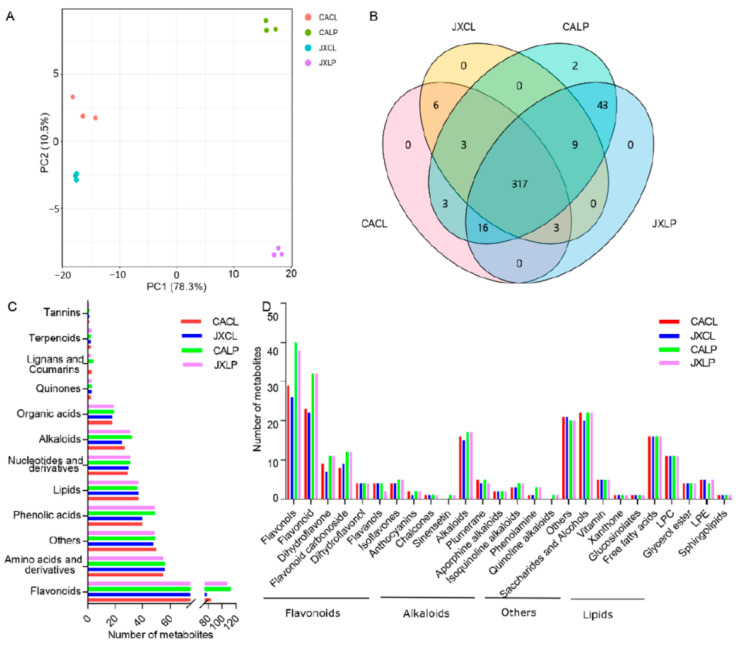
Metabolites detected in CL and LP of two cultivars “CA” and “JX”. (**A**) The principal component analysis of all samples subjected to metabolome. The same color represents replications. (**B**) Venn diagram showing the number of common and differential metabolites identified in CACL, JXCL, CALP, and JXLP. (**C**) The number of metabolites belonging to different categories of class I. (**D**) The category and number of metabolites belonging to flavonoids, alkaloids, others, and lipids. “CA”, China Antique; “JX”, JianXuan-17; CL, cotyledon; LP; lotus plumule.

**Figure 2 molecules-26-00913-f002:**
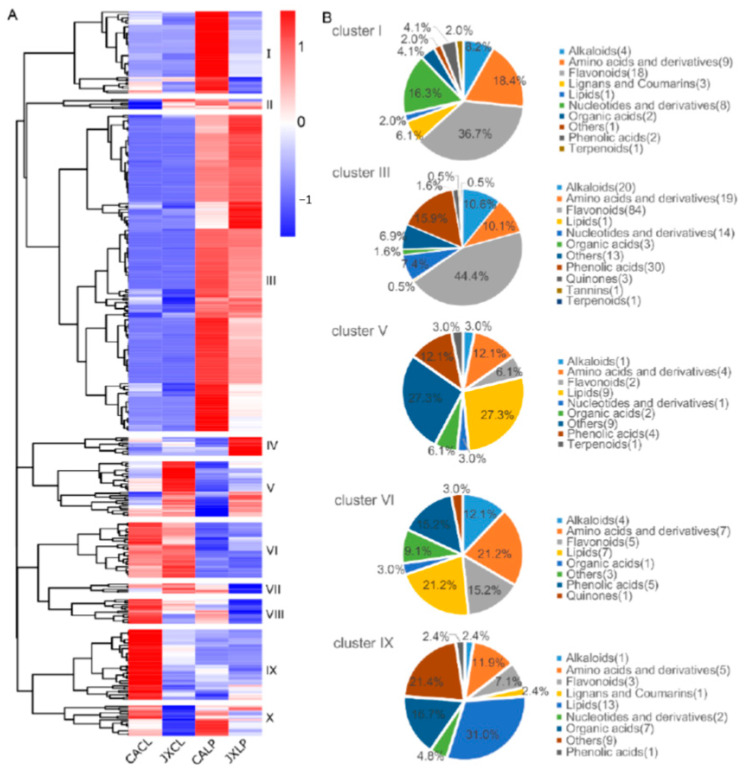
The clustering profiling of the identified metabolites. (**A**) Heatmap indicating the clusters of different metabolites. The red color represents higher content, and the blue color represents the lower content. (**B**) Classification of the metabolites in different clusters in the heatmap. “CA”, China Antique; “JX”, JianXuan-17; CL, cotyledon; LP; lotus plumule.

**Figure 3 molecules-26-00913-f003:**
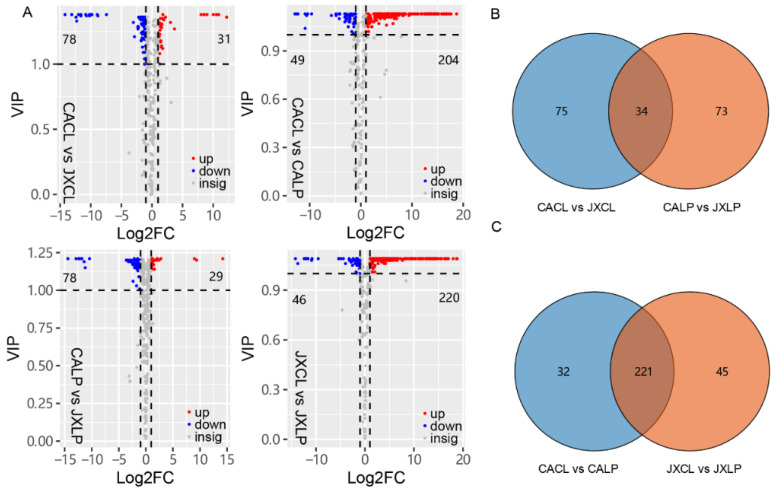
The number of differentially accumulated metabolites (DAMs) in CL and LP of “CA” and “JX”. (**A**) Volcano plots depicting the up- and down- DAMs in the paired comparisons based on the criterions of VIP > 1 and Log_2_FC ≥ 1 or ≤ −1. VIP, variable importance in projection; FC, fold change. (**B**) Venn diagram showing the common or specific DAMs in CL and LP comparing “JX” with “CA”. (**C**) Venn diagram showing the common or specific DAMs in “CA” and “JX” comparing LP with CL. “CA”, China Antique; “JX”, JianXuan-17; CL, cotyledon; LP; lotus plumule; VIP, the value of the variable importance in the project.

**Figure 4 molecules-26-00913-f004:**
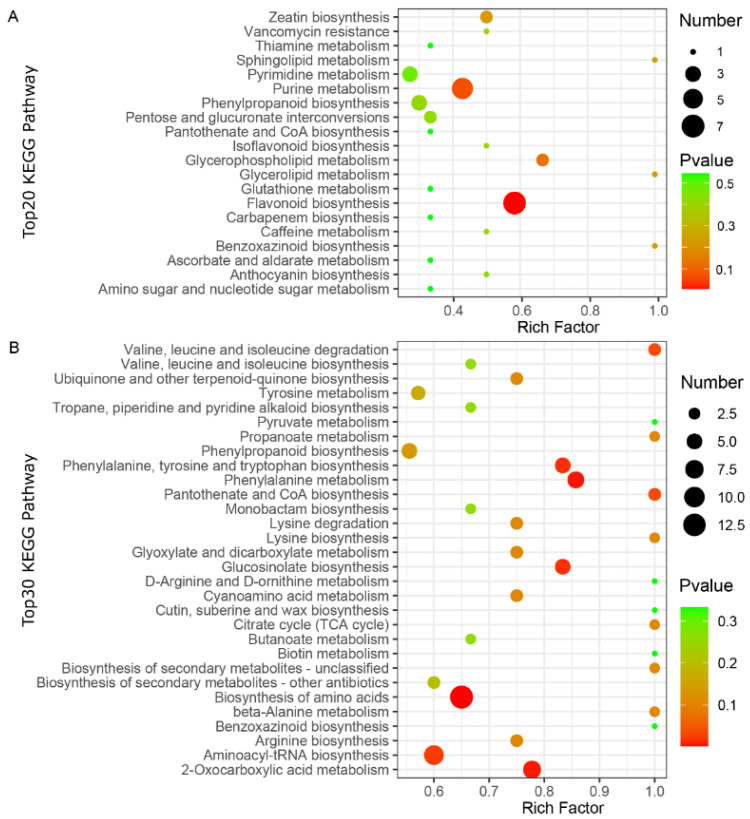
KEGG enrichment analysis of the differentially accumulated metabolites (DAMs) in CL and LP. (**A**) The bubble diagram showing the top20 KEGG pathway enriched for DAMs in LP comparing “JX” with “CA”. (**B**) The bubble diagram showing the top30 KEGG pathway enriched for DAMs in CL comparing “JX” with “CA”. “CA”, China Antique; “JX”, JianXuan-17; CL, cotyledon; LP; lotus plumule; KEGG, Kyoto Encyclopedia of Genes and Genomes.

**Figure 5 molecules-26-00913-f005:**
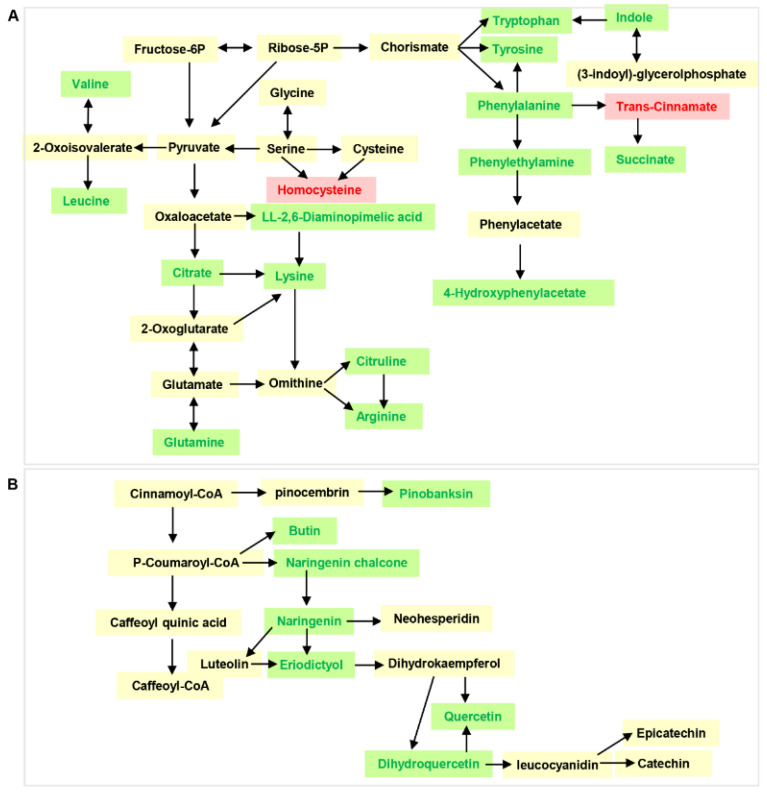
The differentially accumulated metabolites (DAMs) involved in amino acids and flavonoid metabolism. (**A**) The DAMs in CL enriched in biosynthesis of amino acids and related pathway comparing “JX” with “CA”. (**B**) The DAMs in LP enriched in flavonoid biosynthesis pathway comparing “JX” with “CA”. Green color indicated down-accumulated and red color indicated up-accumulated in “JX”. “CA”, China Antique; “JX”, JianXuan-17; CL, cotyledon; LP, lotus plumule.

**Table 1 molecules-26-00913-t001:** Number of differentially accumulated metabolites in different categories in CL and LP of two cultivars “CA” and “JX”. “CA”, China Antique; “JX”, JianXuan-17; CL, cotyledon; LP; lotus plumule.

Category	CACL vs. JXCL		CALP vs. JXLP		CACL vs. CALP		JXCL vs. JXLP	
	Up	Down	Up	Down	Up	Down	Up	Down
Amino acid and derivatives	3	23	3	14	22	11	24	13
Phenolic acids	7	6	2	3	26	5	31	4
Nucleotides and derivatives	1	5	2	10	14	1	13	2
flavonoids	11	17	1	35	103	5	98	7
Quinones	1	0	/	/	/	/	3	1
Lignans and coumarins	0	3	0	4	3	1	2	0
Others	4	7	5	6	9	6	15	8
Tannins	1	0	0	0	1	0	1	0
Alkaloids	3	7	3	3	20	4	23	3
Terpenoids	/	/	1	1	2	1	2	0
Organic acids	0	8	1	2	3	8	5	4
Lipids	0	2	11	0	1	7	3	4

## Data Availability

Data is contained within the article or Appendix A.

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
