# Peer review of "Metabolomics Analyses of Cotyledon and Plumule Showing the Potential Domestic Selection in Lotus Breeding"

_molecules, 2021, doi:10.3390/molecules26040913_

Round 1

Reviewer 1 Report

The authors present a work entitled “Metabolomics analyses of cotyledon and plumule showing the potential domestic selection in lotus breeding”. This work is certainly of interest to readers and touches on a topic of great importance. The metabolomic approach, as a new analytical method, allows to obtain an enrichment of the data on the main classes of bioactive compounds, compared to the previous literature. In general the manuscript is clear, well written and designed and the experiments described correctly. In order to improve the overall quality of the manuscript, I suggest some small changes.

Pg 2 Ln 51-56: “Metabolites are the basis of organism phenotypes, which intuitively and effectively help to understand biological processes and mechanisms[12]. There are about 200,000 metabolites in plants[13-14]. Metabolite profiling has now been performed in many species, such as tomato[15], wheat [16], maize [17], lyceum barbarum[18], loquat[19], sesame[20], which has contributed greatly in gene functional annotation and comprehensive understanding of the cellular response to stresses”, these statements are too general and not useful for discussion. Please, delete them and re-organize the references

Pg 2 Ln 64-67: “The antioxidant activity andconcentration of functional components of lotus seed and rhizome derived from different regions based on high-performance lipid chromatography were investigated[28]. The metabolism-based methods of analyzing multi-pesticide residues have shown that good recoveries for testing pesticides in lotus seeds were obtained [29]. These sentences seem out of context, try to structure this discussion better by including these statements in a more homogeneous and logical way

Pg 2 ln 75-77: “Our results presented here provide valuable information for the metabolite composition in dry lotus seed and new sights in metabolomics-based assisting breeding”. This statement is crucial for defining the aim of the work: please, give more emphasis to this concept, describe if it is the first time that this analysis hjas performed and highlight the relevance of this innovative approach

Pg 8 ln 249: “few studies were focused on…”, please indicate some papers from literature

Pg 9 ln 342: could the authors provide more details about the botanical identification and origin of the botanical material?

Pg 10 ln 353: Please, could the authors report if simple or dynamic maceration was used?

Pg 10 Ln 354: “which was further subjected to centrifugation at 10, 000 g for 10 min.” What was further …”? Please, explain this statement more clearly

Pg 10 Ln 355: please, could the authors explain the utility of the SPE cartridges?

Author Response

Q1: Pg 2 Ln 51-56: “Metabolites are the basis of organism phenotypes, which intuitively and effectively help to understand biological processes and mechanisms [12]. There are about 200,000 metabolites in plants [13-14]. Metabolite profiling has now been performed in many species, such as tomato [15], wheat [16], maize [17], lyceum barbarum [18], loquat [19], sesame [20], which has contributed greatly in gene functional annotation and comprehensive understanding of the cellular response to stresses”, these statements are too general and not useful for discussion. Please, delete them and re-organize the references

R1: Thank you for your comment. We have deleted these content and re-organized the references.

Q2: Pg 2 Ln 64-67: “The antioxidant activity and concentration of functional components of lotus seed and rhizome derived from different regions based on high-performance lipid chromatography were investigated [28]. The metabolism-based methods of analyzing multi-pesticide residues have shown that good recoveries for testing pesticides in lotus seeds were obtained [29]. These sentences seem out of context, try to structure this discussion better by including these statements in a more homogeneous and logical way

R2: Thank you for this suggestion. We have reorganized the statements in the revised manuscript as following: The metabolism-based methods were also applied to analyze the content of functional components and antioxidant activity of lotus seed and rhizome from the different growing region, with varying moisture availability, and pesticide residues in lotus seed [19-20].

Q3: Pg 2 ln 75-77: “Our results presented here provide valuable information for the metabolite composition in dry lotus seed and new sights in metabolomics-based assisting breeding”. This statement is crucial for defining the aim of the work: please, give more emphasis to this concept, describe if it is the first time that this analysis has performed and highlight the relevance of this innovative approach

R3: Thank you for your constructive suggestion. We have revised it based on your comments.

Q4: Pg 8 ln 249: “few studies were focused on…”, please indicate some papers from literature

R4: Thank you for this suggestion. We have added the reference (18, 24,31,40) in the revised manuscript.

Q5: Pg 9 ln 342: could the authors provide more details about the botanical identification and origin of the botanical material?

R5: Thank you for this question. We have added this information in the “Material and methods section”.

Q6: Pg 10 ln 353: Please, could the authors report if simple or dynamic maceration was used?

R6: Thank you. The dissolved samples were refrigerated at 4 °C overnight, during which vortexed 6 times (30 seconds every 30 minutes) to improve the extraction rate. We have revised it in corresponding section.

Q7: Pg 10 Ln 354: “which was further subjected to centrifugation at 10, 000 g for 10 min.” What was further …”? Please, explain this statement more clearly

R7: Thank you. We have re-organized this section in the revised manuscript.

Q8: Pg 10 Ln 355: please, could the authors explain the utility of the SPE cartridges?

R8: Thanks. We have consulted the technical support, and this step was not used in this experiment. I have revised it in the manuscript.

Reviewer 2 Report

Metabolomics provides a comprehensive analysis of primary and secondary metabolites especially in nutritional and medicinal sectors. Metabolite profiling has become very popular in the scientific community, which has been utilized in the conjunction with other omics approaches for a comprehensive understanding of the cellular response to stresses.  In the paper “Metabolomics analyses of cotyledon and plumule showing the m potential domestic selection in lotus breeding”. Authors have performed targeted metabolomics approach based on ultra-performance liquid chromatography-electrospray ionization-tandem mass spectrometry (UPLC-ESI-MS/MS) was utilized to analyze the metabolites in CL and LP of China Antique and Jianxuan. The results presented in the paper provide valuable information for the metabolite composition in dry lotus seed and new sights in metabolomics-based assisting breeding. Metabolism based approach has also been used for analyzing the metabolites composition of different lotus organs.

This work is ready to publish after minor revision. However, I have few comments and I recommend the authors to address. My comments are below.

  • What are the labels on x-axis and y axis of Figure C and D respectively? Add labels to make it clear.
  • What normalization, data scaling and transformation methods etc. were utilized to normalize the data before performing statistical analysis in R and MetaboAnalystR? Please provide details in the statistical analysis method section.
  • How the TCA cycle and glycolytic activities vary across the groups? Please include in the discussion . Central carbon metabolism and glycolysis are the targets of the metabolic changes in the initial growth phase of organs.
  • Figures 1 and 2 are made with low resolution and looks constrained at one corner. Enlarge the figures and provide gap between panels.
  • Supplementary materials-Figures S1-S4, change Figure 1 to Figure S1.

Author Response

Q1: What are the labels on x-axis and y axis of Figure C and D respectively? Add labels to make it clear.

R1: Thank you for your comments. We have added in the revised manuscript.

Q2: What normalization, data scaling and transformation methods etc. were utilized to normalize the data before performing statistical analysis in R and MetaboAnalystR? Please provide details in the statistical analysis method section.

R2: Thank you. We have provided the details based on your comments in the revised manuscript in section of “Statistical analysis”.

Q3: How the TCA cycle and glycolytic activities vary across the groups? Please include in the discussion. Central carbon metabolism and glycolysis are the targets of the metabolic changes in the initial growth phase of organs.

R3: Thank you for your comments. We have re-analyzed the data and found that one metabolite (D-Glucose) involved in glycolysis and 3 metabolites (Fumaric acid, Succinic acid, Citric Acid) involved in the TCA cycle. The detailed discussion has been added in the revised manuscript.

Q4: Figures 1 and 2 are made with low resolution and looks constrained at one corner. Enlarge the figures and provide gap between panels.

R4: Thank you. We prepared each picture with the width of 16 cm, and the smaller pictures in the manuscript maybe caused by layout designer. In the revised manuscript, we have provided the enlarged figures.

Q5: Supplementary materials-Figures S1-S4, change Figure 1 to Figure S1.

R5: Thank you. We have revised it in the revised manuscript.

Reviewer 3 Report

Comments and Suggestions for Authors

The paper entitled “Metabolomics analyses of cotyledon and plumule showing the potential domestic selection in lotus breeding”. The results presented here provide valu-75 able information for the metabolite composition in dry lotus seed and new sights in 76 metabolomics-based assisting breeding.

This work is good. It is mandatory to correct the manuscript in some points:

# line 100-112: -O- should be italic

# line 118: “(37, 9.14%-10.95%),” I suggest put “;” and then should be “(37; 9.14%-10.95%),”

# line 135: “p-” should be italic

# line 185-188: -O- should be italic

# line 269-272: -O- should be italic

# line 280: should be “effect at 200 ug/mL [47].”

# line 306: the authors should put space between “…were decreased” and “[53]”. Please, correct whole paper.

# Materials and Methods: should be “mL” or “uL”. Please, correct whole paper.

# line 409: “p-” should be italic

Author Response

Q1: line 100-112: -O- should be italic

R1: Thank you. We have revised it.

Q2: line 118: “(37, 9.14%-10.95%),” I suggest put “;” and then should be “(37; 9.14%-10.95%),”

R2: Thank you. We have revised it.

Q3: line 135: “p-” should be italic

R3: Thank you. We have revised it.

Q4: line 185-188: -O- should be italic

R4: Thank you. We have revised it.

Q5: line 269-272: -O- should be italic

R5: Thank you. We have revised it.

Q6: line 280: should be “effect at 200 ug/mL [47].”

R6: Thank you. We have revised it.

Q7: line 306: the authors should put space between “…were decreased” and “[53]”. Please, correct whole paper.

R7: Thank you. We have revised it.

Q8: Materials and Methods: should be “mL” or “uL”. Please, correct whole paper.

R8: Thank you. We have revised it.

Q9: line 409: “p-” should be italic

R9: Thank you. We have revised it.